RBI-ThPhys-2024-12

# Magic phase transition and non-local complexity in generalized $W$ State

A. G. Catalano,[1, 2] J. Odavić,[1, 3, 4] G. Torre,[1] A. Hamma,[3, 4] F. Franchini,[1] and S. M. Giampaolo[1]

[1] *Institut Ruđer Bošković, Bijenička cesta 54, 10000 Zagreb, Croatia*
[2] *Université de Strasbourg, 4 Rue Blaise Pascal, 67081 Strasbourg, France*
[3] *Dipartimento di Fisica Ettore Pancini, Università degli Studi di Napoli Federico II, via Cinthia, 80126 Fuorigrotta, Napoli, Italy*
[4] *INFN, Sezione di Napoli*
(Dated: July 1, 2024)

We employ the Stabilizer Rényi Entropy (SRE) to characterize a quantum phase transition that has so far eluded any standard description and can thus now be explained in terms of the interplay between its non-stabilizer properties and entanglement. The transition under consideration separates a region with a unique ground state from one with a degenerate ground state manifold spanned by states with finite and opposite (intensive) momenta. We show that SRE has a jump at the crossing points, while the entanglement entropy remains continuous. Moreover, by leveraging on a Clifford circuit mapping, we connect the observed jump in SRE to that occurring between standard and generalized $W$-states with finite momenta. This mapping allows us to quantify the SRE discontinuity analytically.

Entanglement has played an important role in impetuously developing our understanding of quantum many-body systems [1, 2]. However, over the years it has become increasingly clear that entanglement alone is not able to capture every feature that differentiate quantum from classical systems[3, 4]. The most relevant example of this is the fact that entanglement alone does not guarantees the so-called *Quantum Supremacy* [5]. Indeed, several highly entangled quantum states can be obtained from a fully factorized state by circuits made of Clifford gates [6, 7], i.e. a series of operations that can be efficiently simulated on a classical computer. Quantum advantage is attained at the price of non Clifford resources and exponential increment in the difficulty of simulating a quantum circuit on classical computers [5, 8].

The resource beyond Clifford operations is colloquially known as *magic* [9, 10] and its quantification is a formidable challenge for quantum information science [10]. Recently, a computationally tractable family of measures of Magic for qubit systems called the *Stabilizer Rényi Entropies* (SREs) [11] has been proposed. Indeed, it was proven that SREs with Rényi index greater or equal to 2 are monotones of magic for pure states [12]. Among the others, the one with Rényi index equal to 2 has acquired a prominent role since in some cases is experimentally achievable [13–15].

This family of stabilizer entropies has provided a way for analyzing the complexity of quantum states in quantum many-body systems [22–24]. In gapped local systems, it was found that the SRE of ground states follows a volume law in which the slope can be determined using single-spin expectation values [23]. On the contrary, this local behavior of the SRE disappears in the presence of long-range correlations that can be induced either by placing the system near a phase transition or, as emphasized in recent years, inducing topological frustration in the system [25–28]. SRE, and more generally, other magic measures, have thus proved to be useful in the analysis of quantum many-body phases[9].

In this work, we show that one can employ SRE to characterize a mirror symmetry breaking transition in the anisotropic Heisenberg model that has so far been particularly elusive and hard to detect since, up to today, there are no known non-vanishing order parameters in the thermodynamic limit and where even the analysis of entanglement properties has not been helpful.

The Hamiltonian of the 1D fully anisotropic Heisenberg chain (also known as the XYZ chain) with a global magnetic field along the $z$-axis reads

$$H_{\text{XYZ}} = \sum_{n=1}^{L} \sum_{\alpha} J_\alpha \sigma_n^\alpha \sigma_{n+1}^\alpha + h \sum_{n=1}^{L} \sigma_n^z. \qquad (1)$$

Here and in the following we will always assume that $J_x = 1$ and $|J_y|, |J_z| < J_x$. For certain values of the Hamiltonian parameter this model is exactly solvable, but we will consider it in generality and thus approach it numerically. Within Frustrated Boundary Conditions (FBCs) [44–49], the number of sites is taken to be odd ($L = 2M + 1$ for $M \in \mathbb{N}$) and periodic boundary conditions are assumed ($\sigma_n^\alpha = \sigma_{n+L}^\alpha, \forall n$). FBCs induce a frustration of topological origin in these spin chains, with several interesting consequences on their low-energy properties such as the introduction of an unusual long-range behavior of correlations [48] and the substitution of a gapped system with one in which the ground state is part of band of $2L$ states.

According to [31], the frustrated phase of the XYZ model divides into two regions. In the first, which includes the Ising chain, the ground state is unique and has zero momentum. In [32] it was argued that the line dividing the two regions is a second order boundary QPT, but that analysis did not account for the lattice momentum quantization. In fact, the analysis performed in [31] showed no discontinuities in the free energy and

did not find any observable that survive the thermodynamic limit. In this work, we analyze such a transition, considering it as an example of a general kind of phase transitions that can be characterized in terms of a discontinuity in the SRE. To the best of our knowledge, this is the first instance of a "pure magic transition" in a deterministic quantum system, even if, a shift between local and non-local magic has been recently observed in random quantum circuits [14, 33–35].

The generality of this result comes from the unique behavior of SRE in a family of generalized $W$-states in which each element is defined by its phase. Unlike bipartite entanglement, in the thermodynamic limit, the value of magic remains a function of the phase. As we will show in the following, these states can be mapped into the elements of the lower energy band of the topologically frustrated XYZ model close to the classical point, i.e. when the Hamiltonian is close to the one of classical Ising. Therefore, a crossover in the elements of the lowest energy band is associate to a change for the SRE of the ground state. To extend this result outside the perturbative regime, we generalize the results in [24]. We provide numerical evidence that, in the thermodynamic limit, it is possible to write the SRE of the ground states of a topologically frustrated spin chain as the sum of the SRE of the ground state of the corresponding non-frustrated model plus the SRE of the associated generalized $W$-states. Since without frustration the SRE is continuous within a given phase, the SRE of the topologically frustrated chain shows the same signature of phase transition inherited from the generalized $W$-states even outside the perturbative regime.

We start our analysis by calling to mind the $W$-states [36] that play a pivotal role in quantum information [37–39] finding applications in various quantum protocols, such as anonymous transmission in quantum networks [40], quantum communication [41], and error detection [42]. The family of generalized $W$-states (gWs) that we consider in this letter reads

$$|W_p\rangle = \frac{1}{\sqrt{L}} \sum_{j=1}^{L} e^{\imath p j} \sigma_j^z |-\rangle^{\otimes L}. \qquad (2)$$

Here $L$ is the number of qubits in our system, $p$ is a generic phase, and $|\pm\rangle$ are the eigenstates of the Pauli operator $\sigma^x$ corresponding to eigenvalues $\pm 1$. The original $W$-states are recovered by setting $p = 0$. Being a generalization, some properties of the $W$-states extend to the whole family, while others will depend on $p$. Among the properties of the $W$-states that extend to every gWs there is the entanglement, which is independent of $p$. Indeed, for any bipartition, there are only two non-vanishing eigenvalues of the reduced density matrix and are equal to $(1 \pm x)/2$, where $x$ is the difference between the dimension of the two subsystems normalized by $L$.

For a pure state $|\psi\rangle$ defined in a system of $L$ qubits, the SRE (of index 2) is defined as

$$\mathcal{M}_2(|\psi\rangle) = -\log_2 \left( \frac{1}{2^L} \sum_{\mathcal{P}} \langle \psi | \mathcal{P} | \psi \rangle^4 \right). \qquad (3)$$

Here $\mathcal{P} = \bigotimes_{i=1}^{L} P_i$ runs over all possible Pauli strings, $P_i \in \{\sigma_i^0, \sigma_i^x, \sigma_i^y, \sigma_i^z\}$ with $\sigma_i^0$ represents the identity operator. Although eq. (18) implies a summation of $4^L$ expectation values, it allows for an efficient treatment with tensor networks [16–21].

The expression of the SRE for gWs, Eq.2 is a function on $L$ and $p$ and we evaluated it analytically to be (see Supplementary Material)

$$\mathcal{M}_2(p, L) = -\log_2 \left( -\frac{11 - 12L + \frac{\sin((2-4L)p)}{\sin(2p)}}{2L^3} \right). \qquad (4)$$

In the limit $p \to 0$ we reach the minimum of eq. (23) and recover the SRE for the $W$ state:

$$\mathcal{M}_2(0, L) = 3 \log_2(L) - \log_2(7L - 6). \qquad (5)$$

Moreover, if we require that the states in (2) must be also eigenstates of the translation operator [43] then $p$ becomes the quantized momentum ($p = \frac{2\pi}{L}\ell$) with $\ell$ integer running from 0 to $L - 1$. Within this hypothesis, for $\ell \neq 0$ we have that eq. (23) reduces to

$$\mathcal{M}_2 \left( \frac{2\pi}{L}\ell, L \right)\bigg|_{\ell \neq 0} = \mathcal{M}_2(0, L) + \log_2 \left( \frac{7L - 6}{6L - 6} \right). \qquad (6)$$

Note that this expression is independent from $\ell$, as long as it is finite: the difference $\Delta \mathcal{M}_2(L) = \mathcal{M}_2 \left( \frac{2\pi}{L}\ell, L \right) - \mathcal{M}_2(0, L)$, represents the gap in the SRE, which is associated with the state acquiring non-zero momentum that reduces to $\log_2(7/6)$ in the thermodynamic limit.

The family of states in (2) plays a central role in the study of topologically frustrated 1D systems. Indeed, applying on them the (magic preserving) Clifford circuit

$$\hat{\mathcal{S}} = \prod_{j=1}^{L-1} \mathbf{C}(L, L-j) \left( \prod_{j=1}^{M} \sigma_{2j-1}^z \right) \mathbf{H}(L) \sigma_L^z \prod_{j=1}^{L-1} \mathbf{C}(j, j+1) \Pi^z \quad (7)$$

introduced in Ref. [24] it is possible to obtain the elements of the low energy band of the 1D topologically frustrated XYZ model close to the classical point. Here $\mathrm{H}(j) \equiv \frac{1}{\sqrt{2}}(\sigma_j^x + \sigma_j^z)$ is the *Hadamard gate* on the $j$-th qubit, while $\mathrm{C}(j, l) \equiv \exp\left[\frac{\pi}{4}(1 - \sigma_j^x)(1 - \sigma_l^z)\right]$ is the *CNOT gate* on the $l$-th qubit controlled by the value of the $j$-th one and $\Pi^z = \bigotimes_{j=1}^{L} \sigma_j^z$ is the parity operator along z.

The non-trivial response of AFM spin chains to FBC is also witnessed by an excess of bipartite entanglement beyond the area-law contribution [47, 50, 51]. While these

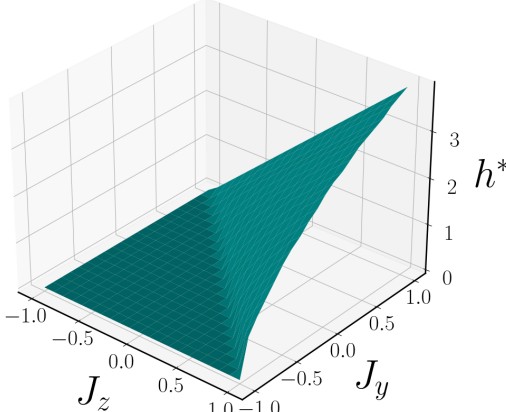

FIG. 1. Value of $h^*$ as function of $J_y$ and $J_z$ ($J_x$ is assumed to be equal to 1) for the Hamiltonian in eq. (1). The data is obtained numerically looking at the momentum of the ground state for a system made of $L = 15$ spins. For $h^* > 0$, choosing $|h| < h^*$ the ground state manifold has dimension equal to 2 and is spanned by states with finite, opposite momenta $p \neq 0$.

properties characterize the whole frustrated phase, accordingly with [31], assuming $J_z \geq -J_y$ there exists a critical value of the external magnetic field $h^* > 0$ such that for $|h| < h^*$ the ground state manifold becomes twofold degenerate and spanned by states with finite, opposite momenta $p \neq 0$. Interestingly, the physics of the whole frustrated phase can be described in a quasi-particle picture through a single delocalized excitation in the ground state of frustrated chains. While in most cases this excitation carries zero momentum, below $h^*$, where the ground-state manifold is at least two-fold degenerate, it owns a non-vanishing one.

In the spirit of adiabatic continuation, let us analyze a particularly simple case where the calculations can be carried out analytically, and subsequently, we will show how the results thus obtained generalize to the entire phase. Therefore, let us focus on the case in which the system is close to the classical point. We define classical point, the case in which $J_x$ is the only non-vanishing Hamiltonian parameter. Indeed, in this case, eq. (1) reduces to a sum of mutually commuting terms, i.e. to a classical Hamiltonian. In this region, exploiting perturbation theory, we obtain that the elements of the lowest energy band can be written as *kink states*

$$|\omega_p\rangle = \frac{1}{\sqrt{2L}} \sum_{k=1}^{L} e^{ipk}(|k\rangle + |k'\rangle), \qquad (8)$$

where $p$ is the quantized momentum, i.e. $p = 2\pi\ell/L$, with $\ell = 0, \dots, L-1$. The kinks are embedded in Neél order states and are made of the union of two extensive sets of states defined as $|k\rangle = \mathcal{T}^k \bigotimes_{j=1}^{M} \sigma_{2j}^z |-\rangle^{\otimes L}$ and $|k'\rangle = \mathcal{T}^k \bigotimes_{j=1}^{M} \sigma_{2j}^z |+\rangle^{\otimes L}$ with $k$ and $k'$ running from 1 to $L$. Turning on $J_y$ $J_z$ and/or $h$ in the proximity of

the classical point we can have either a unique or a two-fold degenerate ground state. In the first case, the lowest energy is obtained by setting $p = 0$, while in the other case, the two ground states display equal but opposite momenta $p$, dependent on the Hamiltonian parameters.

Therefore, it becomes crucial to have a physical quantity able to discriminate between states that yield results dependent on $p$, or at least whether, $p$ is zero or not. In [31] the authors showed how chirality, generally sensitive to the momentum, cannot be used in this case since, even if different from 0 for finite $L$, it vanishes in the thermodynamic limit. A natural second candidate is represented by the entanglement, which has often been used to reveal the presence of non-local correlations. Summarizing the results reported in the supplementary materials, we have that, limiting ourselves to partitions $(A|B)$ composed of connected subsets, regardless of their dimensions, the reduced density matrix $\rho_A(p) = \text{Tr}_B(|\omega_p\rangle\langle\omega_p|)$ admits only 4 non-zero eigenvalues:

$$\lambda_{1,\dots,4} = \frac{1}{4L}\Big(L + 2\gamma\cos(p\chi) \pm \qquad (9)$$
$$\pm \sqrt{(L-2a)^2 + 4L(1 + \gamma\cos(p\chi)) - 4\sin^2(p\chi)}\Big),$$

where $a = \dim(A)$, $\chi = L - a$ and $\gamma = \pm 1$. From eq. (16) it is evident that the momentum dependent contributions scale at most with the inverse square-root of $L$ and thus vanish in the thermodynamic limit. Therefore, for large $L$ the entanglement does not depend on $p$. To provide an example, setting $a = (L-1)/2$ and evaluating the 2-Rényi entropy of $|\omega_p\rangle$ we obtain

$$S_2(\omega_p) = -\log_2\left[\frac{1 + L(4 + L) + 4\cos(p)}{4L^2}\right] \qquad (10)$$

that becomes independent on $p$ when $L$ diverges.

On the contrary, magic works perfectly to detect the finite momentum. Since the states $|\omega_p\rangle$ can be obtained from the $|W_p\rangle$ via a Clifford circuit, they share the same value of magic. When the ground state is unique and carries zero momentum, the value of the SRE is given by eq. (5) but when the ground state acquires a finite momentum it increases by a quantity that stays finite even in the thermodynamic limit.

To extend this result to the whole frustrated phase we note that in the thermodynamic limit it is always possible to write the SRE of the ground state of a topologically frustrated spin chain $|g^{TF}\rangle$ as the sum of the SRE of the ground state of the corresponding non-frustrated model $|g^{NF}\rangle$ plus the SRE of a gWs, i.e.

$$\mathcal{M}_2(|g^{TF}\rangle) = \mathcal{M}_2(|g^{NF}\rangle) + \mathcal{M}_2(|W_p\rangle). \qquad (11)$$

We prove this decomposition numerically. To perform a meaningful finite-size scaling analysis, we need system sizes that go beyond the capabilities of exact diagonalization techniques. Larger chains typically involve an exponential increment in the number of correlation functions

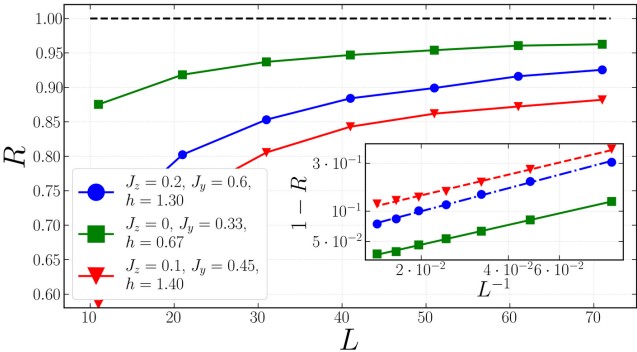

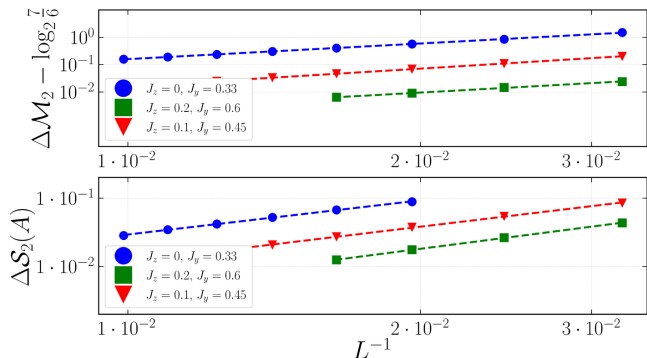

FIG. 2. The ratio $R(p, L)$ defined in eq. (12), as a function of $L$ for different sets of parameters. In both cases, we observe a power-law convergence of $R(p, L) \to 1$ for $L \to \infty$, as highlighted in the inset plot, where we plot $1 - R(p, L)$ as a function of $L^{-1}$ in log-log scale.

FIG. 3. Finite-size scaling analysis of the discontinuity in the SRE (top) and in the entanglement (bottom) for different sets of anisotropies ($J_x = 1$ in all analyzed cases ). Both the two quantities show a power-law decay to the thermodynamic values that are, respectively $\log_2(7/6)$ and 0. The entanglement is evaluated with the 2-Rényi entropy and the data plotted are associated at the partition $(A|B)$ in which $A$ is a connected subsystem made of $(L - 1)/2$ spins.

but, recently, several methods have been introduced to estimate SRE using matrix product states (MPS) representations [17, 18, 21]. In our case the most suited approach is the one proposed in [21] since its worse scaling with the MPS bond dimension, compared for instance to [18], is compensated by the possibility of avoiding any statistical sampling on the distribution of Pauli strings. The latter is problematic since it does not converge easily in the frustrated case, due to the emergence of a multi-peaked distribution for the correlation functions. Therefore, we first compute the chain's ground-state chain in the MPS form using a density matrix renormalization group (DMRG) algorithm [52–54] and then use it to evaluate its SRE. We follow this procedure both to determine the ground state of the topologically frustrated chain $\left|g^{TF}\right\rangle$, and the one of the corresponding non-frustrated model $\left|g^{NF}\right\rangle$, obtained by inverting the signs of $J_x$ and $J_y$.

In Fig. 2 we plotted the quantity

$$R(p, L) = \frac{\mathcal{M}_2\left(\left|\psi^{TF}\right\rangle\right)}{\mathcal{M}_2\left(\left|\psi^{NF}\right\rangle\right) + \mathcal{M}_2 g(|W_p\rangle)}, \qquad (12)$$

that clearly approaches unity as $L \to \infty$ hence proving eq. (11). The data from this analysis would already be sufficient to prove that the phase transition associated with the violation of the mirror symmetry is highlighted by a finite gap of the magic. However, it is also interesting to provide direct verification. Therefore we performed a finite-size scaling analysis of the jump in magic at the transition point $h^*$ for several sets of parameters. To realize this analysis, we fix the values of the anisotropies, determine numerically $h^*$ and plotted the difference in the SRE soon after and soon before this point. In all analyzed cases, the numerical data show a power-law convergence of the amplitude of the discontinuity to the analytically computed value of $\log_2(7/6)$, as shown in Fig. 3. On the contrary, the discontinuity

in the bipartite entanglement shows a power-law convergence to 0, implying that in the thermodynamic limit, it is unable to highlight the presence of the phase transition. This confirms that the SRE witnesses the quantum phase transition associated to the violation of the mirror symmetry in topologically frustrated spin chains, which could therefore be classified as a *first-order magic (SRE) transition*.

Summarizing, we introduced a generalization of $W$-states that promotes a finite momentum. While preserving the value of entanglement of the $W$-states, they possess a greater degree of complexity as highlighted by the SRE. This generalization of $W$-states is extremely relevant in topologically frustrated 1D systems since they can be mapped, through a Clifford circuit into the elements of the lowest energy band close to the classical point. Then, we showed that, since the complexity of the ground states of topologically frustrated chains can be decomposed as the sum of a non-frustrated component and that of the gWs, the transition separating the region of zero momentum ground state from that with finite momenta can be characterized by a discontinuity in SRE. While it was already shown in other works that the SRE can detect measurement-induced phase transitions that are not signaled by the entanglement entropy in quantum circuits, to the best of our knowledge, our result constitutes the first instance of a quantum phase transition that can only be witnessed by the SRE in a deterministic system. The reason behind the fact that only the SRE can capture this quantum phase transition is probably related to the fact that the corrections induced by topological frustration on quantities like the correlation functions typically decay at least as $L^{-1}$, hence vanishing in the thermodynamic limit. The SRE, however, involves the sum of the expectation values of an exponential number of correla-

tion functions and hence can display a finite jump even in the thermodynamic limit. It is important to stress once more that, while the Clifford mapping does not preserve the bipartite entanglement entropy and thus those of g$W$s and the spin ground states differ, they do not show discontinuities when a finite momentum appears.

The nature of this transition, being induced by boundary conditions, has remained controversial so far: the results of this work not only show the first instance of a discontinuity in SRE not accompanied by a similar one in the entanglement in a deterministic model, but further establish complexity in condensed matter/statistical physics systems as a detector of unconventional quantum phase transition. Of course, additional instances of such phenomenology are needed to establish whether complexity is just a proxy to detect transitions (like the entanglement entropy) or if it truly captures something fundamental, like topological order parameters.

*Acknowledgments.—* This work was also supported by the PNRR MUR Project No. PE0000023-NQSTI (J.O. and A.H). A.H. acknowledges financial support PNRR MUR Project No. CN 00000013-ICSC. AGC acknowledges support from the MOQS ITN programme, a European Union's Horizon 2020 research and innovation program under the Marie Skłodowska-Curie grant agreement number 955479.

---

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

**Magic phase transition and non-local complexity in generalized W State – Supplementary Meterials**

**Entanglement Entropies for the $\omega_p$ states for a generic bipartition made of convex subsystems**

As we have seen in the main text, near the classical point the ground state of a topological frustrated system can be well-approximated by a state $\omega_p$ that can be written in the form

$$|\omega_p\rangle = \frac{1}{\sqrt{2L}} \sum_{k=1}^{L} e^{ipk}(|k\rangle + |k'\rangle), \tag{13}$$

Here $p$ is the quantized momentum, i.e. $p = 2\pi\ell/L$, with $\ell = 0, \ldots, L-1$ and $L$ being the (odd) length of the chain. The kinks are embedded in Néel order states and are made of the union of two extensive sets of states defined as

$$\begin{aligned}
\left|k^+\right\rangle &= T^{k-1} \bigotimes_{j=1}^{M} \sigma_{2j}^z |+\rangle^{\otimes L} \\
\left|k^-\right\rangle &= T^{k-1} \bigotimes_{j=1}^{M} \sigma_{2j}^z |-\rangle^{\otimes L} \ .
\end{aligned} \tag{14}$$

In (14) $|\pm\rangle$ denote the eigenstates of $\sigma^x$ associated respectively to the positive/negative eigenvalue in the $x$-direction, $M = (L-1)/2$, while $T$ stands for the translation operator that shifts the state of the system by one single site towards the right. For $k = 1$ the ferromagnetic defect is placed between sites 1 and $L$ while with $k > 1$ the translation operator moves it around the whole chain.

Let us now consider a partition of the system $A|B$ in which both $A$ and $B$ are convex sets, i.e. ensembles of contiguous spins. From (13) we may recover the reduced density matrix obtained by projecting $\omega_p$ into $A$. In the quite general case $a = dim(A) \geq 2$, the reduced density matrix can be written as follows.

$$\rho_A = \mathrm{Tr}_B(|\omega_p\rangle\langle\omega_p|) = \frac{1}{2L}
\begin{pmatrix}
\mathbf{Q}^{(b)} & \mathbf{0}^{(b)} & V^{(b)} & W^{(b)} \\
\mathbf{0}^{(b)} & \mathbf{Q}^{(b)} & W^{(b)} & V^{(b)} \\
(V^{(b)})^\dagger & (W^{(b)})^\dagger & L-b & 2\cos(\chi p) \\
(W^{(b)})^\dagger & (V^{(b)})^\dagger & 2\cos(\chi p) & L-b
\end{pmatrix} \tag{15}$$

In eq. (15) $\rho_A$ is a $2a \times 2a$ square matrix, and we defined $b = a - 1$ and $\chi = L - a$. The reduced density matrix $\rho_A$ is not block diagonal but has a block structure and each one of these blocks has a quite regular structure. To begin, the matrices $\mathbf{0}^{(b)}$ and $\mathbf{Q}^{(b)}$ are both $b \times b$ square matrices. All the elements of the first are zeros, i.e. $\mathbf{0}_{m,n}^{(b)} = 0 \, \forall m, n$, while the elements of $\mathbf{Q}^{(b)}$ obey to the following law $\mathbf{Q}_{m,n}^{(b)} = \exp(-\imath(m-n)p)$. On the contrary, both $V^{(b)}$ and $W^{(b)}$ are column vectors made of $b$ rows and one single column. The $n$-th element of $V^{(b)}$ can be written as $V_n^{(b)} = \exp(-\imath(b+1-n)p)$, while for $W^{(b)}$ we have $W_n^{(b)} = \exp(-\imath(L-n)p)$

Diagonalizing this matrix with the help of Mathematica, and testing the results so obtained with a purely numerical code, we have found that all the eigenvalues are zero except four. These four non-vanishing eigenvalues can be put in the form

$$\lambda_{1,\ldots,4} = \frac{1}{4L}\left(L + 2\gamma\cos(p\chi) \pm \sqrt{(L-2a)^2 + 4L(1+\gamma\cos(p\chi)) - 4\sin^2(p\chi)}\right), \tag{16}$$

where $\gamma$ is a dicotomic real parameter of modulus 1, i.e. $\gamma = \pm 1$. The four eigenvalues are recovered considering all the possible combinations of the $\pm$ sign in front of the square root and the values of $\gamma$. From this expression, all the different entropic measures of the entanglement can be easily recovered. A case in which the analytical expression of the entanglement entropy becomes very easy is the 2-Rényi entropy that, after some steps, can be reduced to

$$S_2(a,p) = -\log_2\left[\frac{L(2+L) - 2a(L-a) + 2\cos(p\chi)}{2L^2}\right] \tag{17}$$

from which, setting $a = (L-1)/2$, we can recover the result presented in the main text

## Analytic derivations of Stabilizer Rényi entropy for a $W_P$ state

The main result shown in the main part of the letter is based on the behavior of the SRE on the family of $W_P$ states, which is a family that generalizes the well-known $W$ state. To evaluate the SRE, let us start by recalling its expression for a generic pure state $|\psi\rangle$. It reads

$$\mathcal{M}_2(|\psi\rangle) = -\log_2\left(\frac{1}{2^L}\sum_{\mathcal{P}}\langle\psi|\,\mathcal{P}\,|\psi\rangle^4\right), \tag{18}$$

where the sum runs over all possible Pauli strings $\mathcal{P}$ that can be defined on the system. Taking into account that any $|W_p\rangle$ state can be written as

$$|W_p\rangle = \frac{1}{\sqrt{L}}\sum_{j=1}^{L}e^{\imath pj}\sigma_j^z\,|-\rangle^{\otimes L}, \tag{19}$$

we immediately see that to determine the SRE on $|W_p\rangle$, we have to evaluate the terms

$$O(\mathcal{P}) = \sum_{j,k=1}^{L}\exp[\imath(j-k)p]\,\langle-|^{\otimes L}\,\sigma_k^z\mathcal{P}\sigma_j^z\,|-\rangle^{\otimes L} \tag{20}$$

It is easy to see that in the large majority of cases $O(\mathcal{P}) = 0$, but with two important exceptions. The first exception is when the Pauli string is made only by the identity and $\sigma^x$ operators, i.e. when $\mathcal{P}$ becomes $\mathcal{P}' = \bigotimes_{k=1}^{L}\sigma_k^\alpha$, where $\alpha \in \{0, x\}$. In this case, the absolute value of each $O_{j,k}(\mathcal{P}')$ depends on the number $l = 0,\ldots,L$ of $\sigma_k^x$ operators in the string, and it is equal to $\|\frac{L-2l}{L}\|$. Taking into account all the possible combinations of identity and $\sigma^x$ operators, the contribution of these terms becomes

$$\sum_{\mathcal{P}'}O(\mathcal{P}') = \sum_{l=0}^{L}\left(\frac{L-2l}{L}\right)^4\frac{L!}{l!(L-l)!}. \tag{21}$$

The second exception is represented by the Pauli strings with only two operators in the set $\{\sigma^y, \sigma^z\}$. Within this hypothesis, the Pauli string can be written as $\mathcal{P}'' = \bigotimes_{k=1,k\neq i,j}^{L}\sigma_k^\alpha \otimes (\sigma_j^\beta\sigma_k^\gamma)$ where $\alpha = 0, x$ while $\beta, \gamma = y, z$. This contribution comes from the fact that such strings are able to shift the $|+\rangle$ from the site $j$ to the site $k$ and vice versa. When $\beta = \gamma$ both these two terms have the same sign, so giving a contribution proportional to $\cos[(j-k)p]/2L$. On the contrary, when $\beta \neq \gamma$ they show opposite signs, so contributing proportional to $\sin[(j-k)p]/2L$ Naming $r = j-k$ and summing over all possible Pauli strings we have

$$\sum_{\mathcal{P}''}O(\mathcal{P}'') = L\sum_{l=0}^{L-2}\sum_{r=1}^{L-1}\frac{(L-2)!}{l!(L-2-l)!}\left[\left(\frac{2\cos(pr)}{L}\right)^4+\left(\frac{2\sin(pr)}{L}\right)^4\right]. \tag{22}$$

Putting the two non-vanishing contributions together in the definition of the SRE of order 2 we recover, after some steps, the following expression

$$\mathcal{M}_2(W_p) = -\log_2\left(-\frac{11-12L+\frac{\sin((2-4L)p)}{\sin(2p)}}{2L^3}\right). \tag{23}$$