# Peer review of "Magic phase transition and non-local complexity in generalized W State"

_SciPost Physics_

## Round 2 · Referee Report · Anonymous (Referee 1) · 2025-6-4

Strengths
- This work discovers the first instance where nonstabilizerness can detect a phase transition, when entanglement fails.
Weaknesses
- Analysis is too simplistic
- The general utility of the finding is not clear
Report
This work studies the nonstabilizerness, quantified by the stabilizer Renyi entropy (SRE), in topologically frustrated systems, which can be mapped to the generalized W state in some parameter regime. Through this mapping, the authors show that the SRE displays a jump at a quantum phase transition associated to mirror symmetry breaking. This behaviour is in contrast to the entanglement which remains continuous. As such, this is the first instance of a transition which can only be detected by the SRE (however, see below). These findings were numerically illustrated in the 1D XYZ model with frustrated boundary condition.
While the finding itself is interesting, I am not convinced with its general utility. The authors have mentioned the momentum operator as another possible probe of the transition, which is however only well-defined in the presence of translational invariance (TI). The authors thus argued that the SRE is potentially more useful for general systems. However, from the analysis of the generalized W state, it appears that the jump in the SRE is inherently connected to the finite momenta, where the momenta is only well-defined with TI. It is thus uncler if such a transition in SRE also appears in systems without TI. The present analytical and numerical analysis concern a system with TI, where the momentum operator should also work, and easier to compute than the SRE. Therefore, the authors still cannot claim that the transition can only be detected by the SRE. In order to corroborate the generality of the SRE, the authors should provide some examples of systems without TI, and potentially provide numerical analysis on those systems. For example, can the TI in the present model be broken somehow in a way that preserves the phases and their transitions?
I do not think that this work provides a novel link between different research areas, since the connection between nonstabilizerness and frustrated systems as well as the W state has been discussed previously by some of the authors in their previous Scipost publication (Ref. [24]). It may still open a new pathway if the concern above has been appropriately addressed, as it is currently unclear if the SRE really has potential for general use beyond other quantities as the authors claim.
Other comments: - In Fig. (3), the authors show the SRE difference just before and after a point, which is somewhat unphysical. This difference is connected to the derivative of the SRE, which should display discontinuity. I suggest the authors to also show the derivative, which has previously been showed to be an indicator of a transition (e.g. Ref. [21]) - The discontinuity in SRE appears in the subleading term, which is typically nontrivial to extract. Can the authors comment on whether this term can be extracted through specific linear combinations? - Furthermore, can the authors comment on the universality of this term? - Minor: the abbreviation "AFM" is not defined
Requested changes
- Clarify the generality of the SRE in systems without translational invariance
- Clarify the nature of the subleading term and its universality
Recommendation
Ask for major revision
We thank the referee for her/his careful reading of our manuscript and for the beneficial criticisms. We appreciate the opportunity to clarify our perspective regarding the universality of the stabilizer Rényi entropy (SRE) as a diagnostic tool. As per her/his suggestion, we have: - included a new appendix (Appendix C) where we have presented analysis of a system derived from the one in Eq. (7) but where translation invariance is broken explicitly. - In the third section, we have included a paragraph to make remarks about the generality of our approach
Hereafter, we present a point-by-point response to each of the issues mentioned below.
----------------------------------+
The referee is correct to note that, for translationally invariant systems, the momentum operator naturally probes the transition. As we discuss in the rewritten manuscript, however, the SRE remains when translation invariance is explicitly broken. To demonstrate that, we have extended our analysis to a locally defective analog of the XY model with broken translation symmetry. The computations, presented in Appendix C, show that while the flaw qualitatively alters the nature of the transition—increasing the ground-state degeneracy below the critical point and replacing it with a dense succession of crossovers—the SRE still has a discontinuity at the critical point. This establishes that the SRE is not just associated with the presence of a well-defined momentum quantum number, but instead establishes more significant alterations to the ground-state structure.
We respectfully disagree, therefore, with concern that the transition seen by the SRE is exclusive to translationally invariant systems.
Although the momentum operator is undoubtedly simpler to compute under the models of the main text, its ubiquity does not transfer outside this setup, as it is no longer a well-defined observable when translational symmetry is lost. The SRE, by contrast, is always well-defined and continues to signal the phase transition even when translational symmetry is lost. 2) We thank the referee for this comment and the opportunity to clarify the statement regarding the originality of our work. It is indeed true that in our previous paper (Ref.~[24]) we have already addressed the role of SRE in topological frustration systems, where it provided valuable insights into the connection between nonstabilizerness and frustration and its correspondence with generalized $W$ states.
However, the present manuscript does make a significant improvement. Here we extend the analysis to topologically frustrated systems in which there is a transition involving a change from a non-degenerate to a two-fold degenerate ground state. We show that the SRE has a sharp discontinuity through the transition, even when more traditional diagnostics such as the entanglement entropy are smooth. This new situation highlights the power of the SRE to locate singular quantum phase transitions that cannot be accessed by usual tools. 3) We would like to thank the referee for her/his useful remark. In our example, however, the derivative of the SRE is not a good detector of the transition. This is because the phenomenon has nothing to do with a classical continuous evolution of the ground state but with a crossover from one qualitatively different regime to another: a single ground state and a two-fold manifold formed by states with opposite, non-zero momenta.
Thus, as a consequence, the derivative of the SRE is smooth at the transition, while the discontinuity appears explicitly in the SRE itself. Thus, we particularly focused on highlighting the SRE difference before and after the transition point, which perfectly captures the abrupt change in ground-state structure. See also ref. [29] 4) Thanks to the referee for calling our attention to this fundamental observation. As we detail in the updated manuscript, the discontinuity of the SRE is in fact due to the contribution at subleading order, but not the leading term that is simply extensive and linear in $L$. More generally, and at least in the short-range one-dimensional case, we have that the SRE and other extensive quantum resources can be separated in the thermodynamic limit into two terms: a leading extensive term and a subleading one connected with the topological structure of the ground-state manifold.
Therefore, an investigation of quantities such as that in Eq.(13) can allow us to distinguish between the subleading term and the leading term, which we also demonstrate in the amended text. For what the referee asks in the second half of his question, while our present results indicate that this subleading term contains universal information regarding the ground-state topology, a full characterization of its universality class is not within the scope of this paper. We consider this an interesting direction for further research and plan to follow it up in detail in the future.

Author: Salvatore Marco Giampaolo on 2025-09-23 [id 5851]
(in reply to Report 2 on 2025-06-09)Warnings issued while processing user-supplied markup:
Add "#coerce:reST" or "#coerce:plain" as the first line of your text to force reStructuredText or no markup.
You may also contact the helpdesk if the formatting is incorrect and you are unable to edit your text.
We would like to thank the referee for careful reading of our manuscript and for the helpful comments and suggestions. We found reports very helpful in refining the quality and scope of our work, and we have revised the text accordingly.
Following his suggestions we have: - inserted the sections in order to make our work more readable; - inserted a new appendix (Appendix C) in which we have presented the analysis of a system deriving from the one in Eq. (7) but in which the translation invariance is explicitly violated.
In the following, we provide a point-by-point response to each of the points raised below.
1) We thank the referee for the valuable comment. In fact, in translationally invariant systems, the momentum operator is naturally a marker of the transition. However, as discussed in the new paper, its definition breaks down at once as soon as we give up translation invariance. Thus, we focused on the SRE, which is still well-defined even in non-translationally invariant situations. To satisfy the referee's demand, we explicitly tested this robustness by examining a model with a local defect that breaks translation symmetry. The results, presented in Appendix C, are that while the defect qualitatively alters the nature of the transition, removing the ground-state degeneracy below the critical point, the SRE still shows a sharp discontinuity, therefore invariably detecting the phase transition. This serves to highlight the usefulness of using the SRE rather than the momentum operator as a more general diagnostic.
Also, the results in Appendix A illustrate that, in the thermodynamic limit, any observable expressible as a finite sum of Pauli strings necessarily fails to detect the transition. Thus, even if one were to posit different observables might in practice be able to in fact detect it, their calculation will likely be of comparable computational cost as for the SRE.
2) We thank the referee for bringing this fascinating and nontrivial point to our attention. It is easy to observe from Eq.(2) that the generalized W states differ from the standard version by relative phases in the coefficients. Since participation entropies depend solely on the moduli of these coefficients, we would naively expect them to be insensitive to the phase transition considered here. This image may be changed, however, if one looks at the system in Eq.(7), or rather its translation-invariant counterpart. There, then, the calculation of the participation entropies exactly is only feasible with information on all the diagonal elements of the reduced density matrix, which restricts calculable thinking to relatively modest system sizes. This limitation can be addressed by the employment of new methods like those newly presented in Kožić & Torre (2025) (https://arxiv.org/abs/2502.06956), and is thus an interesting direction for future work.
3) We thank the referee for her/his remark. In this matter, however, we should like to clarify our position respectfully. In the revised manuscript, we report evidence that the SRE may detect the transition not only when there is spatial translation invariance but even when it is explicitly broken. Since the SRE is an actual observer of quantum magic, we don't think that investigating other quantities which are extremely closely related would contribute very much to the physical picture already in the paper. All the same, we fully agree with the referee that exploring complementarities in diagnostics is of potential interest. Here, a line of research that is extremely promising is the exploration of quantities different from, yet still conceptually related to magic, such as the nonlocal magic. We have begun a first-principles study along these lines already; technically the research on nonlocal magic is extremely challenging, though, and our current results, for extremely small system sizes, are not robust enough yet to make decisive predictions about its behavior across the entire transition. For that reason, we believe it is better to leave this vital question to future research, where it can be treated in the extent it merits.

---

## Round 2 · Referee Report · Anonymous (Referee 2) · 2025-6-9

Report
-My main concern is that the authors' arguments do not convincingly justify characterizing the studied transition as a 'magic phase transition'. In fact:
1) As also the authors write "it is conceivable that other combinations of [Pauli] strings, with different physical meanings, could detect this transition. The most natural example that comes to mind for our case is the momentum operator". So wouldn't it be simpler, and wouldn't it have a clearer physical meaning, to use the momentum operator to detect the transition? I understand that momentum is not well-defined in non-translationally invariant systems; however, the only model studied here (Eq. 7) is translationally invariant. In fact, it could be valuable to include results for such a scenario, for instance by considering a disordered version of Eq. 7.
In general, is the use of SRE to detect what essentially amounts to the emergence of excitations with finite and opposite (intensive) momenta somewhat overstated? Can simpler observables capture the same phenomenon?
2) Since the SREs are a sort of 'participation entropy' (in the Pauli basis), I would actually be interested in knowing whether other participation entropies can also detect the studied transition. I am referring, for example, to participation entropies in the computational basis, i.e., the inverse participation ratios. Are you aware if that is the case?
3) In general, it seems to me that the authors do not provide sufficient arguments to support the claim that SREs are in any way special compared to a generic combination of Pauli strings. Moreover, if one intends to refer to the transition as a magic transition, it might be useful to investigate the behavior of other magic quantifiers, beyond the SREs. I am referring, for example, to the robustness of magic, or similar monotones (even though these are limited to small system sizes).
-The text is at times difficult to follow, occasionally resembling a stream of consciousness rather than a structured argument. I recommend that the authors reconsider the overall structure of the manuscript and, at the very least, introduce a clear division into sections. This currently missing element is essential to improve the readability. So for instance: Introduction, Model, Results, Conclusions, etc.
-I am not sure I understood Eq.8 and the next sentences. The symbol \mathcal{T} is undefined, so it is very difficult to understand why Eq.8 represents kink states.
Requested changes
1) Introduce Sections and improve readability 2) Provide more compelling evidence to support the claim of a "magic phase transition".
Recommendation
Ask for major revision

---

## Editorial Decision

unknown